# Comparative Analysis of Inhibitory and Activating Immune Checkpoints PD-1, PD-L1, CD28, and CD86 in Non-Melanoma Skin Cancer

**DOI:** 10.3390/cells13181569

**Published:** 2024-09-18

**Authors:** Linus Winter, Jutta Ries, Christoph Vogl, Leah Trumet, Carol Immanuel Geppert, Rainer Lutz, Marco Kesting, Manuel Weber

**Affiliations:** 1Department of Oral and Cranio-Maxillofacial Surgery, Friedrich-Alexander-Universität Erlangen-Nürnberg (FAU), 91054 Erlangen, Germany; linus.winter@uk-erlangen.de (L.W.); christoph.vogl@uk-erlangen.de (C.V.);; 2Deutsches Zentrum Immuntherapie (DZI), Friedrich-Alexander-Universität Erlangen-Nürnberg (FAU), 91054 Erlangen, Germany; 3Comprehensive Cancer Center Erlangen-EMN (CCC ER-EMN), 91054 Erlangen, Germany; 4Comprehensive Cancer Center Alliance WERA (CCC WERA), 91052 Erlangen, Germany; 5Bavarian Cancer Research Center (BZKF), 91052 Erlangen, Germany; 6Department of Operative Dentistry and Periodontology, Friedrich-Alexander-Universität Erlangen-Nürnberg (FAU), 91054 Erlangen, Germany; 7Institute of Pathology, University Hospital Erlangen, Friedrich-Alexander-Universität Erlangen-Nürnberg (FAU), 91054 Erlangen, Germany

**Keywords:** NMSC, TMA, IHC, TME, PD-1, PD-L1, CD28, CD86, cSCC, BCC

## Abstract

The establishment of immunotherapy applying immune checkpoint inhibitors (ICI) has provided an important new option for the treatment of solid malignant diseases. However, different tumor entities show dramatically different responses to this therapy. BCC responds worse to anti-PD-1 ICIs as compared to cSCC. Differential immune checkpoint expression could explain this discrepancy and, therefore, the aim of this study was to analyze activating and inhibitory immune checkpoints in cSCC and BCC tissues. Tissue microarrays of the invasive front as well as the tumor core of BCC and cSCC samples were used to evaluate PD-1, PD-L1, CD28, and CD86 expression and their topographic distribution profiles by chromogenic immunohistochemistry. QuPath was used to determine the labeling index. The expression of PD-1, PD-L1, and CD28 was significantly higher in both the tumor core and the invasive front of cSCC samples as compared to BCC (*p* < 0.001). In addition, the ratios of PD-L1/CD86 (*p* < 0.001) and CD28/CD86 (*p* < 0.001) were significantly higher in cSCC. The invasive front of both tumor entities showed higher expression levels of all immune markers compared to the tumor core in both tumor entities. The significantly higher expression of PD-1, PD-L1, and CD28 in cSCC, along with the predominance of the inhibitory ligand PD-L1 as compared to the activating CD86 in cSCC, provide a potential explanation for the better objective response rates to anti-PD-1 immunotherapy as compared to BCC. Furthermore, the predominant site of interaction between the immune system and the tumor was within the invasive front in both tumor types.

## 1. Introduction

Non-melanoma skin cancer (NMSC) is the most commonly diagnosed cancer in North America, Australia/New Zealand, and Europe [1]. In 2018, more than 1 million patients were diagnosed with this type of cancer worldwide, with an estimated death toll of more than 65,000 [1]. However, due to the fact that basal cell carcinoma (BCC) and cutaneous squamous cell carcinoma (cSCC) are not recorded by most cancer registries [2] the true number may be much higher, wherein both entities account for approximately 93% of NMSC cases [3].

Furthermore, NMSC cases have shown a steady increase of about 2% (2014–2019) per year [4] and the reasons are manifold. Probably the most common causes for the development of BCC and cSCC are the prevailing trend towards tanned skin [4], increased life expectancy [5], and improved screening as part of early detection programs [6]. 

In addition, patients treated with immunosuppressive drugs have been shown to be at higher risk of developing tumors such as NMSC [7,8,9,10,11,12,13]. Solid organ transplant recipients have been observed to have up to a 65-fold increased risk of developing cSCC after kidney or heart transplantation [14], and a 10-fold increased incidence of developing BCC [15]. These facts further indicate that the adaptive immune response is important in BCC and possibly even more so in cSCC [16,17]. 

Therapeutic options for both cSCC and BCC range from standard surgical resection to cryosurgery, topical immune activation therapy with imiquimod, radiation therapy, and others [18,19,20,21]. Although there is no single standard of care for NMSC patients, guidelines exist for the treatment of cSCC and BCC [21,22,23,24]. For both BCC and cSCC, the preferred treatment remains complete surgery with removal of all tumor tissue (R0) [21,22,23,24]. However, systemic therapy may be indicated for locally advanced or metastatic NMSC [25]. The clinical benefit of immune checkpoint inhibitors (ICIs) has led to Food and Drug Administration (FDA) approval for IT in cSCC and BCC [25]. In metastatic or locally advanced cSCC, anti-PD-1 therapy has been successfully used with objective response rates of approximately 50% [26]. A partial response was defined as a decrease in the sum of target lesion diameters of at least 30% [26]. More recently, cemiplimab, an anti-PD-1 antibody, was approved by the FDA for use in locally advanced and metastatic BCC (laBCC/mBCC) with complete response rates of 6% and an objective response by ICR of 31% in laBCC after hedgehog inhibitor therapy [16]. In addition, cSCC has been reported to respond more vigorously and quickly to immunotherapy (IT) than BCC [27].

It has been shown that the efficacy of immunotherapy significantly depends on the mutational burden present, with some tumor entities performing better or worse than would be predicted by the mutational burden [28]. Both BCC and cSCC have some of the highest mutational burdens of any human cancer [29,30]. Applying the Yarchoan et al. formula [28] to the respective mutational burdens yields an estimated objective response rate of 41.1% for BCC and 40.5% for cSCC. Hence, the comparison of estimated and reported objective response rates implies that factors in addition to mutational burden play a role in the efficacy of anti-PD-1-IT, as cSCC shows higher response rates. 

Activating and inhibitory immune checkpoint receptors and their ligands play a central role in tumor immunology [31,32]. The PD-1-PD-L1 immune checkpoint pathway is one of the most relevant inhibitory signals for T cells. In contrast, activation of the CD28 receptor with its ligand CD86 can stimulate the immune system as it is an important driver of T-cell immune responses. The T-cell-bound CD28 receptor was first discovered in the early 1980s, where it was found to play a critical role in T-cell activation. The ligands CD80 (B7 type 1 protein) and CD86 (B7 type 2 protein) are expressed on antigen-presenting cells (APCs) and bind to the same receptors CD28 and CTLA-4 [33]. The interactions of CD80 or CD86 with CD28 promote T-cell survival, proliferation, and cytokine production on both effector and regulatory T cells [34]. Since tumor-infiltrating lymphocytes (TILs) and macrophages can be detected in the tumor tissue of patients with cSCC [35,36,37] and BCC [35], elucidation of CD28 and CD86 expression within the TME may be crucial. This could improve the understanding of IT success using ICI therapy for patients with NMSC, in addition to the analysis of PD-1 and PD-L1 expression in both BCC and cSCC.

Therefore, the aim of this study is to analyze the TME of cSCC and BCC by tissue microarray (TMA) immunohistochemistry (IHC) for PD-1, PD-L1, CD28, and CD86 expression in the invasive front as well as in the core of both tumor entities. Furthermore, the composition of the TME will be investigated by correlation analysis of expression patterns of different marker pairings and by comparison of expression ratios. We hypothesize that differences in TME composition may provide a plausible explanation as to why BCC and cSCC respond differently to IT with ICI, and further that topographic differences in expression patterns may explain the difference in clinicopathological behaviors of between BCC and cSCC.

## 2. Materials and Methods

### 2.1. Patients Collective

Ethical approval was granted by the local ethics committee of the Friedrich-Alexander University Erlangen-Nuremberg (Case 54_17Bc) and the study was conducted in accordance with the Declaration of Helsinki. In total, the cohort consisted of 93 BCC and 108 cSCC cases presenting in 55 females and 98 males, with several patients presenting with multiple BCC/cSCC. Demographics and histomorphological data of the BCC and cSCC cases are summarized in Table 1.

### 2.2. Sampling of Tissue

#### 2.2.1. Tissue Sampling

All patients were treated at the Department of Oral and Maxillofacial Surgery, University Hospital Erlangen. Cases presented to the department between 2010 and 2020 were included. Histopathologic reports provided by the Institute of Pathology were reviewed for case selection. In addition, all tissue samples were examined by light microscopy (Axio Imager A2, Carl Zeiss, Jena, Germany) to ensure high sample quality. In each paraffin-embedded tumor sample, 1.5 mm wide circular areas were determined to be located either in the tumor core or in the invasive margin of the tumor. If possible, due to the size of the tumor sample, three 1.5 mm cores were then defined and punched for TMA generation. If all three cores were available after staining and digitalization, the mean value of all three cores was calculated. TMAs were then constructed using the TMA Grand Master (3D-Histech, Budapest, Hungary). Tissue cores were divided by tumor entity as well as by location (invasive front vs. tumor core) for independent TMA. Thus, a single TMA represented both a tumor entity and one of the two localization attributes. Next, 2 µm sections were cut from each TMA using a rotary microtome (HistoCore AUTOCUT, Leica Biosystems, Nussloch, Germany). During the sectioning, some cores loss was observed among the TMAs. Sections were then prepared for immunohistochemical (IHC) staining.

#### 2.2.2. Immunohistochemical Staining

Sections were deparaffinized in xylene (Carl Roth GmbH & Co KG, Karlsruhe; Germany). The sections were subsequently rehydrated by means of a descending alcohol (Carl Roth GmbH & Co KG, Karlsruhe; Germany) series. At no time after deparaffinization were sections exposed to a non-submerged condition to avoid desiccation and the risk of high background due to non-specific antibody binding. Epitope unmasking was performed by heat-induced epitope retrieval. In order to do so, samples were heated in citrate buffer (pH 6.0; PD-L1) for 30 min at 99 °C or in EDTA buffer (pH 9.0; PD-1, CD28, CD86) for 20 min at 99 °C and allowed to rest at room temperature. Immunohistochemical staining was performed using the polymer detection method with an automated IHC stainer (Autostainer Plus, Dako, Agilent, Santa Clara, CA, USA). IHC staining was performed as previously described by [38,39]. As primary antibodies, PD-1 (AB137132; Lot: GR284234-11; Abcam, Cambridge, UK; dilution 1:500), CD28 (AB243228; Lot: GR3277376-2; Abcam, Cambridge, UK; dilution 1:250), PD-L1 (PI516C01; Lot: Z777; DCS, Hamburg, Germany; dilution 1:200), and CD86 (AB234000; Lot: GR3278421-7; Abcam, Cambridge, UK; dilution 1:100) were used. In addition, for visualization, the Histofine Simple Stain MAX PO staining kit (DAB kit, medac, Wedel, Germany) in combination with the CD28 primary antibody and the BrightVision detection system (HRP kit, medac, Wedel, Germany) was used in combination with the PD-1/PD-L1/CD86 primary antibodies according to medac’s instructions. Human tonsil and lymph node tissue were stained as positive controls in each run. Human oral mucosa stained with antibody diluent was used as a negative control. During the staining, a certain amount of core loss was observed. The highest was observed in CD28. Representative TMA cores of each staining are shown in Figure 1.

#### 2.2.3. Digitization and Statistical Analysis

IHC-stained sections were scanned for digitization in cooperation with the Digital Pathology Core Unit of the Institute of Pathology, University Hospital Erlangen (Pannoramic 250 Flash III, 3D Histech, Budapest, Hungary). Digital quantification and analysis were performed using QuPath-0.2.3 (Queen’s University Belfast, Belfast, Northern Ireland, UK), an open-source bioimage analysis software [40]. TMA grids were applied to each slide according to the initial TMA setup, allowing for unique core allocation. Individual tissue cores were set to Missing = True if cores were missing, the scan was out of focus, or the tissue showed artifacts and was, therefore, not quantifiable or suitable for further analysis. The stain vectors were optimized using the program’s Estimate Stain Vector command. Brightness and contrast were adjusted individually for each scan and channel, as scanned slides vary slightly in initial contrast and brightness values. A classifier was then trained for each TMA slide. Positive cell detection was performed with optimized values for each slide to ensure optimal cell detection and differentiation. The trained classifier afterwards automatically classified the remaining labeled tumor and stroma into the following cells: PD-1, CD28, PD-L1, and CD86 positive immune or tumor cells and PD-1, CD28, PD-L1, and CD86 negative immune or tumor cells using the RandomTrees algorithm. Analysis was performed in the epithelial tumor compartment and the tumor stroma as shown in Figure 2. Automated cell counting of positive and negative cells was then performed over the entire available tissue area. The data were analyzed by relating the total number of positive cells to the number of cells detected in a given compartment as stroma or total cell labeling index (LI). The stroma labeling index was calculated by dividing the number of positively stained stromal cells by the total number of stromal cells. The total cell labeling index was calculated by dividing the total number of positively stained cells by the total number of all cells. Box plots show median, interquartile range, and minimum (min) and maximum (max) values. Two-sided adjusted *p*-values ≤ 0.05 were considered statistically significant. *p*-values ≤ 0.001 were considered highly statistically significant. The data analysis was performed using the Mann–Whitney U test with IBM SPSS Statistics version 24 (Released 2016, IBM SPSS Statistics for Windows, IBM Corp., Armonk, NY, USA). Ratios of marker pairings were analyzed by dividing their respective total cell labeling indices for BCC and cSCC. In addition, PD-1/CD28/PD-L1/CD86 expressions were correlated separately for BCC and cSCC. Correlation analysis was performed using Spearman’s Rho correlation test. *p*-values were judged statistically significant at either the *p* ≤ 0.05 (*), or *p* ≤ 0.01 (**) level. Spearman correlation coefficient values and two-sided adjusted *p*-values were calculated for BCC and cSCC, respectively. For *p*, moderate positive correlation (*ρ* = 0.40 to 0.59) and strong positive correlation (*ρ* = 0.60 to 0.79) were attributed.

## 3. Results

### 3.1. Clinicopathologic Results

Gender distribution was similar in both BCC cases (33 females, 60 males) and cSCC cases (36 females, 72 males). The age distribution of BCC (73.7 ± 13.4 years) and cSCC (80.7 ± 11.02 years) was also relatively close. Cases in which tumor (T) node (N) metastasis (M) staging (TNM staging) was performed were in accordance with the 2017 TNM classification. Most cases presented as T1 carcinoma (BCC n = 45, 48.4%; cSCC n = 48, 41.7%). None of the BCC cases presented with lymph node metastases (pN+), while nine cSCC cases (8.3%) showed nodal involvement (pN+). No BCC cases had distant metastases, while two cSCC cases (1.9%) did. Histologic grading was not available for BCC cases. However, among cSCC cases, 22 were graded as G1 (20.4%), 44 were graded as G2 (40.7%), and 39 were graded as G3 (36.1%). Detailed information on the demographics and histomorphological parameters of the cohort is shown in Table 1.

### 3.2. Distribution of Immune Cell Infiltration in Different Tumor Compartments

#### 3.2.1. Expression of PD-1, CD28, PD-L1, and CD86 in the Invasive Front and Tumor Core in BCC and cSCC

The PD-1 total cell LI count was significantly higher in the invasive front and the tumor core compartment of cSCC as compared to BCC (*p* < 0.001; Table 2). The PD-1 stroma-positive cells (stromal LI) were significantly more abundant in both compartments for cSCC (*p* < 0.001; Table 2). The CD28 total cell LI showed significantly higher expression within the invasive front (*p* = 0.005; Table 2) and the tumor core of cSCC cases (*p* < 0.001; Table 2) as compared to BCC cases. The CD28 stromal LI showed a statistically significant difference in the invasive front (*p* = 0.002; Table 2), as well as in the tumor core (*p* = 0.001; Table 2), with higher values in cSCC than in BCC cases. The PD-L1 positive cells were observed in the tumor and more frequently in the stromal cells. The PD-L1 total cell LI showed statistically significantly higher levels in the invasive front of cSCC (*p* < 0.001; Table 2), as well as in the tumor core (*p* < 0.001; Table 2), favoring cSCC over BCC. The PD-L1 stromal LI were significantly higher in the invasive front (*p* = 0.001; Table 2) and tumor core (*p* < 0.001; Table 2) of cSCC cases than in BCC cases. The CD86 total cell LI values were significantly higher in both the invasive front (*p* < 0.001; Table 2) and tumor core (*p* < 0.001; Table 2) of BCC cases as compared to cSCC cases. The CD86 stromal LI showed significantly higher levels in both the invasive front (*p* < 0.001; Table 2) and the tumor core (*p* < 0.001; Table 2) in BCC as compared to cSCC. Additional results are available in the online article as Appendix A.

In summary, all antibodies except CD86 showed significantly higher levels in cSCC as compared to BCC for both the invasive front and the tumor core. Only the CD86 expression was significantly higher in BCC as compared to cSCC for both compartments.

#### 3.2.2. Differences in Expression Ratios of PD-1/PD-L1, CD28/CD86, PD-1/CD28, and PD-L1/CD86 in BCC and cSCC by Total Cell LI

The ratio of PD-1/PD-L1 was significantly higher in the invasive front (*p* < 0.001; Figure 3a) as well as in the tumor core (*p* < 0.001; Figure 3b) in favor of BCC as compared to cSCC. Significantly higher ratios were also found in the ratios of CD28/CD86 and PD-L1/CD86 between cSCC and BCC in both the invasive front (*p* < 0.001; Figure 3c,g) and the core (*p* < 0.001; Figure 3d,h) in favor of cSCC. Finally, the ratio of PD-1/CD28 was significantly higher in BCC as compared to cSCC tumor cores (*p* = 0.028; Figure 3f). Additional results are available as Appendix A accompanying the online article. 

In summary, PD-L1 dominates over PD-1, whereas CD28 dominates over CD86, and, finally, the inhibitory PD-L1 dominates over the activating CD86 in cSCC as compared to BCC. 

#### 3.2.3. Comparison of Expression Patterns in the Invasive Front and Tumor Core of PD-1, CD28, PD-L1, and CD86 in BCC and cSCC by Total Cell LI

Differences in expression patterns by tumor compartment were compared within BCC and cSCC by total cell LI. For PD-1, significant differences were found in both BCC (*p* < 0.001; Figure 4a) and cSCC (*p* < 0.001; Figure 4b), with higher levels in the invasive front as compared to the tumor core. For CD28, higher expression levels were found in BCC (*p* < 0.001; Figure 4c) in favor of the invasive front. PD-L1 showed significant differences when comparing the expression patterns, with higher values in cSCC (*p* < 0.001; Figure 4f) in favor of the invasive front. Finally, CD86 showed a significant difference in expression by compartment only in cSCC (*p* = 0.002; Figure 4h), with higher levels observed in the invasive front. Additional results are shown in Figure 4 and as Appendix A accompanying the online article.

In summary, regardless of the tumor entity, all significant results showed an increase in expression within the invasive front as compared to the tumor core.

#### 3.2.4. Spearman Correlation of PD-1, CD28, PD-L1, and CD86 in BCC and cSCC by Total Cell LI

Spearman correlation for PD-1, CD28, PD-L1 and CD86 was performed for both BCC and cSCC by total cell LI. In BCC, a moderate positive correlation (*ρ* = 0.40 to 0.59) was obtained for PD-1/PD-L1 (*ρ* = 0.496, *p* < 0.001) and PD-L1/CD86 (*ρ* = 0.522, *p* < 0.001). A strong correlation (*ρ* = 0.60 to 0.79) was achieved for PD-1/CD86 (*ρ* = 0.636, *p* < 0.001; Figure 5c). In cSCC, a moderate correlation was achieved for PD-1/CD28 (*ρ* = 0.472, *p* = 0.001; Figure 5b).

## 4. Discussion

In continuation of our group’s previous work on immune cell infiltration in BCC and cSCC [35], the aim of this study was to make a quantitative analysis of the inhibitory and activating immune checkpoint pathways PD-1/PD-L1 and CD28/CD86 in both BCC and cSCC. Although both BCC and cSCC have some of the highest mutational burdens among solid malignancies [29,30,41], which suggests an excellent response rate to anti-PD-1 therapy [28], the reported objective response rates to immunotherapy fell short of expectations. Therefore, other factors of the tumor-associated immune infiltrate are likely to be responsible for the observed reduced therapeutic efficacy of anti-PD-1-IT, particularly in BCC.

Our results highlight the facts that PD-1, PD-L1, and CD28 expression was significantly higher in cSCC as compared to BCC for both the invasive front and the tumor core. Only CD86 expression was significantly higher in BCC as compared to cSCC for both the tumor core and the invasive front compartments. 

PD-1 expression in tumor-associated macrophages has been observed to have a negative effect on phagocytic potency against tumor cells [42]. However, PD-1/-PD-L1 blockade can lead to increased macrophage phagocytosis [42] and, therefore, appears to have a macrophage-related beneficial effect in IT [35]. Furthermore, Harada et al. and Vilain et al. reported that tumor-promoting M2-polarized macrophages are associated with increased PD-L1 expression in solid tumors and may, therefore, serve as a prognostic marker for immune checkpoint therapy [43,44]. As we were able to demonstrate relevant expression of PD-L1 in both BCC and cSCC, this finding would be consistent with our previously published results reporting CD68 expression in both BCC and cSCC, with significantly higher expression levels in the latter [35]. 

Regarding the TME, PD-L1 signaling, through interaction with its receptor PD-1, has been linked to inhibition of T-cell activation and proliferation [45,46], resulting in a tumor-promoting immune response. The PD-1-PD-L1 pathway is also a central regulator of T-cell exhaustion in cancer, which can be overcome by anti-PD-1 immunotherapy [47]. 

In contrast, binding of CD28 to its ligand CD86 leads to T-cell activation with increased T-cell survival [48], which is antagonized by the inhibitory and checkpoint receptors cytotoxic T-lymphocyte-associated protein 4 (CTLA4) and PD-1 [49]. Hall et al. hypothesized that BCC has a reduced antigen presentation as compared to cSCC, which may be due in part to an increased number of Treg cells [50]. Treg cells show a selective preference for CD86 over CD80 to provide co-stimulation via CD28, subsequently leading to increased Treg-cell proliferation and survival [51]. As CD86 expression was significantly higher in BCC, this may help to explain the apparent ability of BCC to evade immune surveillance. In addition, CD8+/PD-1+/CD28− T cells from non-small cell lung cancer (NSCLC) patients showed a lower response to anti-PD-1 treatment as compared to CD8+/PD-1+/CD28+ T cells [52]. Kim et al. further postulated that CD28 is a marker of anti-PD-1 antibody responsiveness [52]. This may also be true in NMSC and may further explain the better objective response rates observed in cSCC as compared to BCC, as CD28 expression was significantly higher in cSCC. However, further research is needed to potentially establish CD28 as a predictive marker of anti-PD-1 IT response in NMSC.

The role of mutational burden, regarding its significant effect on objective response rates to anti-PD-1 IT, was highlighted in recent findings [28]. In this regard, a high mutational burden is associated with higher levels of tumor neoantigens, which, in turn, may be a target for the immune system [53]. Since BCC has a marginally higher mutational burden as compared to cSCC [41], the question arises as to why BCC response rates to PD-1 inhibition are less effective. The significantly higher expression of PD-1, PD-L1, and CD28 in cSCC discovered in the current study may provide the missing link to this question, with its insight into the NMSC TME. Furthermore, other factors may alter PD-L1 expression in BCC and, therefore, possibly its response to IT, as a recent study found that arsenic exposure changed levels of PD-L1 expression in BCC patients [54].

To our knowledge, this is the first study that describes the expression of activating and inhibitory immune checkpoints in terms of their topography in both cSCC and BBC and provides a novel point of view on another part of the topographic contexture at the invasive front and tumor core of both entities. To enable the analysis of the topographic contexture, the tumor tissue was divided into two compartments to allow the comparison of the expression patterns. First, the tumor core, and second, the invasive front, which represents the transition zone from the tumor to the surrounding healthy tissue. In each compartment, an analysis of the entire tissue as well as an isolated assessment of the tumor stroma was performed. In BCC, all markers showed increased expression in the invasive front as compared to the core, with significant differences for PD-1 and CD28. In cSCC, PD-1, PD-L1, and CD86 showed a significant increase of expression within the invasive front as compared to the tumor core, in contrast to CD28, which did not. Within our samples, we were able to show that most markers are expressed in the invasive front, suggesting that tumor–host interaction appears to be most prominent at the site of the invasive front. In addition, previous findings of our group showed that the highest infiltration of CD8 and CD68 cells is located in the invasive front of both BCC and cSCC patients [35]. These findings may highlight their potential for immunogenic activity. In colorectal cancer, NSCLC, melanoma, and head and neck cancer, subsets of T cells are present in the core and at the invasive margin [55]. In colorectal cancer, the proportion of primary tumors with high infiltrates of CD4 and CD8 T cells, particularly in the tumor core, is lower in patients with recurrent tumors [55]. In contrast, high infiltrates of CD3 and CD8 T cells observed in the tumor and invasive margin are favorable predictors of recurrence time and overall survival [56]. In NSCLC, an immune checkpoint immunoscore calculated on the basis of variables around CD8 and PD-L1 densities and localization was considered as a powerful tool for predicting the efficacy of ICI in patients with NSCLC [57]. In conclusion, the localization of biomarkers varies among tumors, and further research is needed in NMSC, to find better predictive models regarding survival and IT efficacy in BCC and cSCC patients.

To gain a deeper insight into the immune cell context of PD-1, CD28, PD-L1, and CD86 within the TME, the amount of positively stained cells was used to obtain ratios, then used for comparison between BCC and cSCC. In BCC, the PD-1/PD-L1 ratio was significantly higher than in cSCC. On that account, there is a greater abundance of PD-L1 in relation to the ligand receptor PD-1 in cSCC. Since PD-L1 can be found on a variety of cells such as tumor cells, T cells, macrophages, and others [58], the ratio of PD1/PD-L1 may have different effects on antitumor immunity. In our study, the expression of PD-L1 was observed in tumor cells and stromal cells. The number of PD-L1-expressing cells was higher in the stroma as compared to the epithelial tumor compartment; however, a further subclassification of stromal PD-L1-expressing cells was not possible for methodological reasons. Nevertheless, in advanced melanoma, for example, a higher PD-L1 expression of all expressing cells is associated with an improved ICI IT response [59]. Thus, further research on the relevance of PD-1/PD-L1 expression in BCC and cSCC, as well as further subclassification and identification of PD-L1 expressing cells, is needed to fully understand their association with the anti-PD-1 IT response. 

Furthermore, our results show that the ratio of PD-L1/CD86 is significantly higher in cSCC as compared to BCC. This suggests that the inhibitory PD-L1 predominates over the activating ligand CD86 in cSCC, whereas the activating CD86 predominates over the inhibitory PD-L1 in BCC. In hepatocellular cancer, patients with high expression of PD-L1 or low expression of CD86 had a poor prognosis. Furthermore, PD-L1 and CD86 may have a specific role as potential biomarkers and novel therapeutic targets for prognostic assessment [60]. This may also be true for NMSC, as high expression of PD-L1 correlates with poor differentiation of cSCC tumor tissue [61]. However, further research is needed to establish PD-L1 and CD86 as prognostic biomarkers in both BCC and cSCC. Our previously published results showed no significant differences in CD8/CD68 ratios between BCC and cSCC [35]. Although CD8 expression is mainly observed on the membranes of cytotoxic T cells, the expression levels of T cells as compared to that of macrophages seem to be the same in BCC and cSCC. However, the predominance of CD86 labeling over CD28 LI in BCC may indicate the presence of more APCs as compared to T cells. In cSCC, the opposite can be observed, as CD28 expression is more prevalent than CD86. APCs express pattern recognition receptors such as Toll-like receptors [62], which can be activated by imiquimod [63]. In a case report by Dika et al., three patients with in situ cSCC presented with cSCC recurrence after imiquimod treatment [64]. In contrast, imiquimod was approved by the FDA for the treatment of superficial BCC in 1999 [65] and was shown to be effective in most cases of superficial BCC [66]. Furthermore, MHC-I is barely expressed in BCC unless treated with imiquimod, unlike cSCC, which shows MHC-I expression [67]. This may further elucidate how BCC seems to evade immune surveillance and how immune surveillance can be improved by IT.

### Limitations of the Study

Within the study design, the use of TMA for IHC staining provided an opportunity to stain many patient samples on a single slide. Therefore, only representative areas of the entire tissue sample were analyzed, rather than the entirety of a cross-sectional sample area as compared to whole slide analysis. In addition, a monoplex chromogenic IHC assay was used. Thus, the comparison of different expression patterns was performed on the same patient tissue sample, but not on the same cross-sectional sample area itself. The use of consecutive TMA slides, where possible, may have reduced the effect of spatial dispersion to a large extent, but may, nonetheless, lead to a slight deviation in pattern expression due to an altered localization within the sample. For each marker, not all cases could be included in the analyzation, as core loss was observed during slide sectioning and staining. Further, due to the use of one TMA slide per marker, the lack of tissue abundance limited the analysis for CD28 of the full study cohort. Further, to draw more elaborate conclusions, healthy tissue samples from patients could have been analyzed for a direct comparison of patients, as well as accompanying associations with clinicopathological characteristics and the patient’s prognosis. To mitigate these limitations in the future, the work group will seek to improve future study designs and to establish multiplex immunofluorescent approaches.

## 5. Conclusions

The significant upregulation of PD-1, PD-L1, and CD28, as well as the calculated ratio of increased PD-L1 inhibitory ligand to T-cell activating ligand CD86, in cSCC as compared to BCC, provides a possible explanation for the better objective response rates to anti-PD-1 therapy in cSCC as compared to BCC. Furthermore, our results show that for both BCC and cSCC, the tumor–host interaction of the investigated biomarkers seems to be most pronounced at the invasive front. Finally, we observed that in cSCC, the inhibitory PD-L1 predominates over the activating CD86, whereas in BCC, the role was reversed. However, further research is needed to gain insight into the localization and distance between different cell subtypes by chromogenic multiplex IHC or immunofluorescence multiplex IHC in NMSC to improve the understanding of the respective TMEs of BCC and cSCC.

## Figures and Tables

**Figure 1 cells-13-01569-f001:**
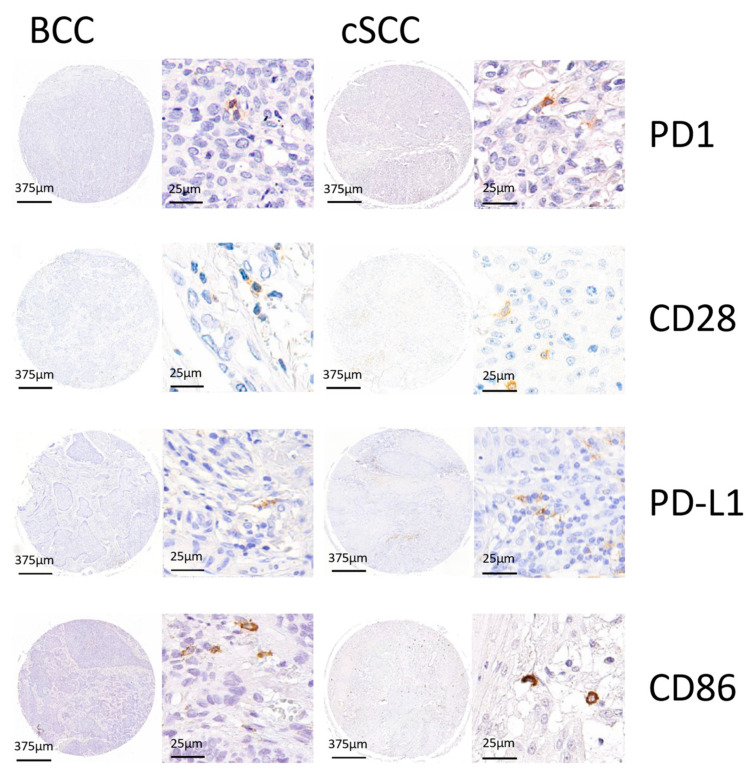
Representative TMA cores of PD-1, CD28, PD-L1, and CD86 for basal cell carcinoma (BCC) and cutaneous squamous cell carcinoma (cSCC) are shown to showcase the different IHC stainings with hematoxylin and DAB signal.

**Figure 2 cells-13-01569-f002:**
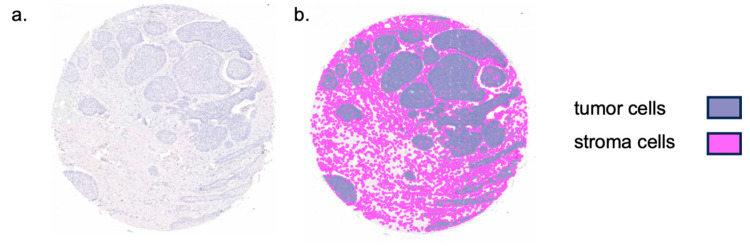
Representation of TMA core tissue segmentation. (**a**) Sample tissue microarray (TMA) core of basal cell carcinoma. (**b**) Colored tissue segmentation overlaying the TMA core. Tissue was separated into tumor and stromal tissue as displayed.

**Figure 3 cells-13-01569-f003:**
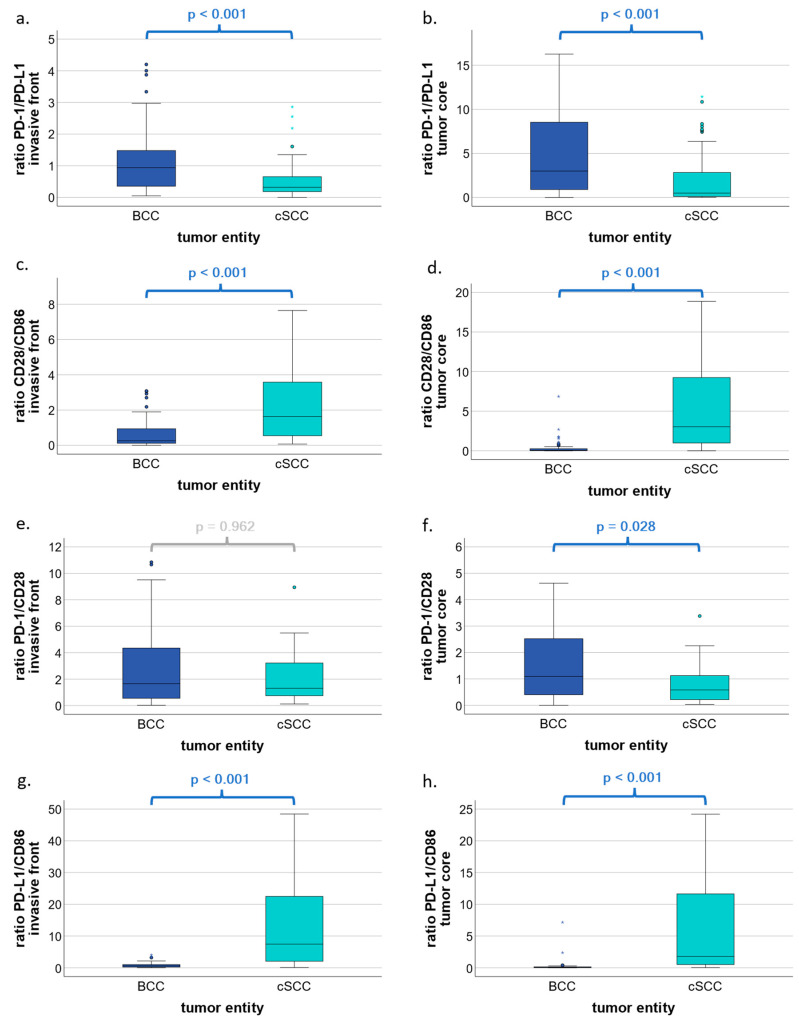
Ratios of PD-1/PD-L1, CD28/CD86, PD-1/CD28, and PD-L1/CD86 in BCC and cSCC by total cell labeling indices. Box plots show ratios of PD-1/PD-L1, CD28/CD86, PD-1/CD28, and PD-L1/CD86 in basal cell carcinoma (BCC) and cutaneous squamous cell carcinoma (cSCC). (**a**,**b**) Box plots represent the ratio of PD-1-expressing cells to PD-L1-expressing cells with respect to the invasive front and the tumor core, respectively. (**c**,**d**) Boxplots represent ratios of CD28-expressing cells to CD86-expressing cells with respect to the invasive front and the tumor core, respectively. (**e**,**f**) Boxplots represent ratios of PD-1-expressing cells to CD28-expressing cells with respect to the invasive front and the tumor core, respectively. (**g**,**h**) Boxplots represent ratios of PD-L1-expressing cells to CD86-expressing cells with respect to the invasive front and the tumor core, respectively. *p* values were calculated using the Mann–Whitney U test. * represent extreme outliers.

**Figure 4 cells-13-01569-f004:**
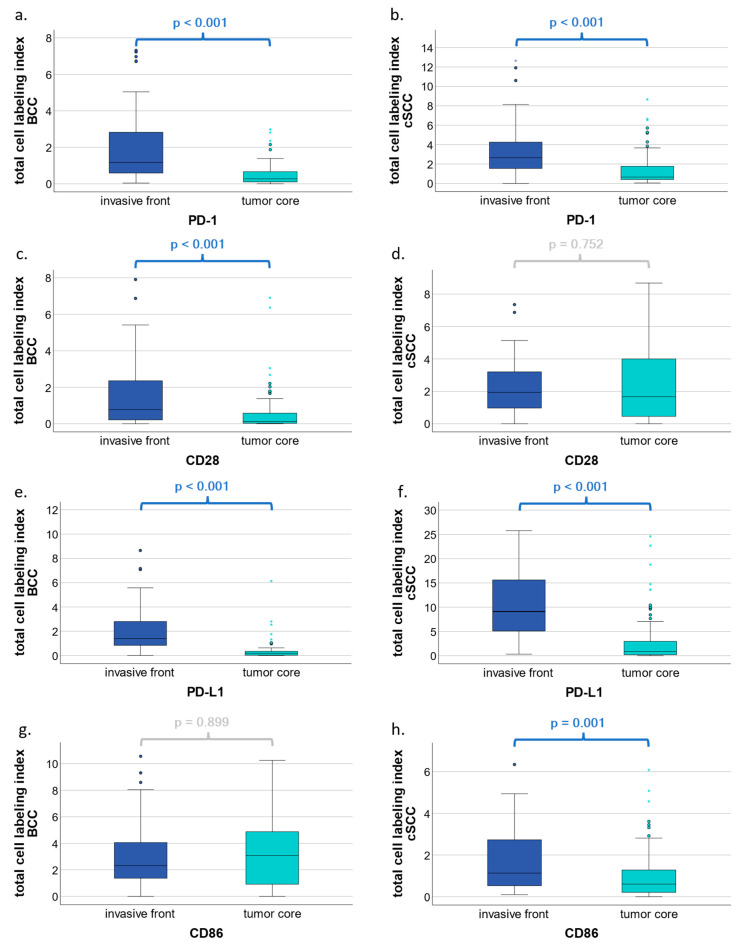
Comparison of expression patterns in the invasive front and tumor core of PD-1, CD28, PD-L1, and CD86 in BCC and cSCC by total cell labeling indices. Box plots show a comparison of marker expression in the invasive front versus the tumor core of PD-1, CD28, PD-L1, and CD86 in basal cell carcinoma (BCC) and cutaneous squamous cell carcinoma (cSCC) (**a**–**h**). The tumor core is composed of mostly epithelial tissue with some stromal tissue, whereas the invasive front represents the transition zone from tumor to stromal tissue. Total cell labeling indices include cell counts of both tumor epithelial and stromal cells within the invasive front or tumor core. *p* values were calculated using the Mann–Whitney U test. * represent extreme outliers.

**Figure 5 cells-13-01569-f005:**
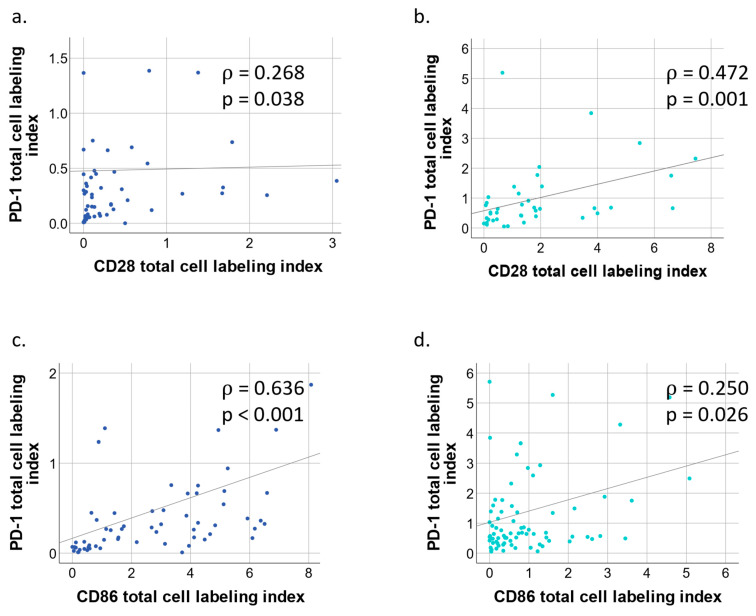
Simple scatter plot of BCC/cSCC tumor core PD-1/CD28 and PD-1/CD86 by total cell labeling indices. (**a**) Scatter plot of basal cell carcinoma (BCC) with fit line of PD-1 by CD28 total cell labeling indices. Correlation coefficient *ρ* = 0.268, *p* = 0.038. (**b**) Scatter plot of squamous cell carcinoma (cSCC) with fit line of PD-1 by CD28 labeling indices. Spearman correlation coefficient *ρ* = 0.472, *p* = 0.001. (**c**) Scatter plot of basal cell carcinoma (BCC) with fit line of PD-1 by CD86 total cell labeling indices. Correlation coefficient *ρ* = 0.636, *p* < 0.001. (**d**) Scatter plot of squamous cell carcinoma (cSCC) with fit line of PD-1 by CD86 total cell labeling indices. Spearman correlation coefficient *ρ* = 0.250, *p* = 0.026.

**Table 1 cells-13-01569-t001:** Description of patient collective.

		BCC	cSCC
total number of patients	women	55	
	men	98	
total number of cases		93	108
	women	33	36
	men	60	72
mean age [years]	women	76.5 ± 13.8	82.9 ± 12.79
	men	70.9 ± 13.87	78.4 ± 8.92
pT	T1	45	48
	T2	4	23
	T3	2	22
	T4	2	1
	TX	40	14
pN	N0	16	43
	N1	0	4
	N2	0	3
	N3	0	2
	NX	77	56
pM	M0	16	65
	M1	0	2
	MX	77	41
grading	G1	0	22
	G2	0	44
	G3	0	39
	GX	93	3
thickness [mm]	mean	1.4	1.1
	max	6.0	4.1
	min	0.1	0.1
infiltration depth [mm]	mean	1.0	0.7
	max	7.0	3.0
	min	0.04	0.10

Values represent the number of cases by n in basal cell carcinoma (BCC) and cutaneous squamous cell carcinoma (cSCC). TX, NX, MX, and GX represent cases without classification.

**Table 2 cells-13-01569-t002:** Comparison of PD-1, CD28, PD-L1, and CD86 means, SD, and *p*-values by labeling indices.

		Invasive Front Stroma Labeling Index		Invasive Front Total Cell Labeling Index		Tumor Core Stroma Labeling Index		Tumor Core Total Cell Labeling Index	
Group/Marker		BCC	cSCC	BCC	cSCC	BCC	cSCC	BCC	cSCC
PD-1	n	69	50	69	50	69	85	69	85
	mean	2.38	4.92	1.91	4.00	0.81	2.15	0.50	1.42
	SD	2.43	5.23	2.04	4.82	0.89	2.23	0.65	1.69
	*p*-value	***p* < 0.001**		***p* < 0.001**		***p* < 0.001**		***p* < 0.001**	
CD28	n	64	47	64	47	65	54	65	54
	mean	2.76	3.89	2.21	2.73	0.99	4.82	0.88	3.50
	SD	4.14	4.22	3.50	3.08	1.88	6.4	2.22	4.98
	*p*-value	***p* = 0.002**		***p* = 0.005**		***p* < 0.001**		***p* < 0.001**	
PD-L1	n	64	48	64	48	72	103	72	103
	mean	2.64	4.91	1.93	11.90	0.45	5.04	0.26	5.20
	SD	2.02	4.93	1.63	10.24	1.02	10.04	0.39	10.69
	*p*-value	***p* = 0.001**		***p* < 0.001**		***p* < 0.001**		***p* < 0.001**	
CD86	n	66	45	66	45	71	95	71	95
	mean	4.55	2.39	3.28	1.66	6.42	1.71	3.18	0.98
	SD	3.68	1.95	3.06	1.47	4.81	2.54	2.49	1.15
	*p*-value	***p* < 0.001**		***p* < 0.001**		***p* < 0.001**		***p* < 0.001**	

Values represent the number of cases (n), median, standard deviation (SD), and *p*-value (Mann–Whitney U test) of marker expression in basal cell carcinoma (BCC) and cutaneous squamous cell carcinoma (cSCC). Here, stromal labeling indices include only cell counts obtained from stromal cells within the invasive front or tumor core, whereas total cell labeling indices include cell counts from both tumor epithelial and stromal cells within the invasive front or tumor core. Significant (*p* ≤ 0.05) and highly significant (*p* ≤ 0.001) values are highlighted in bold.

## Data Availability

The raw data supporting the conclusions of this article will be made available by the authors without undue reservation.

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
