# Peer review of "Comparative Analysis of Inhibitory and Activating Immune Checkpoints PD-1, PD-L1, CD28, and CD86 in Non-Melanoma Skin Cancer"

_cells, 2024, doi:10.3390/cells13181569_

Round 1
Reviewer 1 Report
Comments and Suggestions for Authors
General comments: This is a methodically correct study comparing the expression of 4 immune checkpoints in two non-melanoma skin cancer types, basal cell carcinoma and cutaneous squamous cell carcinoma. However, many questions arise with regard to the interpretation and presentation of the results.
Specific comments
Major problems:
- The starting point of the study does not seem entirely convincing. The authors state that BCC responds significantly worse to anti-PD-1 ICIs than cSCC, however, the available efficacy results are based on few trials in the case of BCC, furthermore, the dose regimen was different in the case of the two cancer types, which makes it difficult to compare the results (no side by side comparisons were made), so any “significant” difference cannot be stated.
- The amount of novel results provided by the study is limited. Potential associations of the studied parameters with clinicopathological characteristics or with patients’ prognosis were not evaluated, only BCCs and cSCCs were compared. However, it is already known from the authors’ previous study (reference 44) that cSCCs seem more immunogenic, being more heavily infiltrated by CD8+ T cells and CD68+ macrophages compared to BCCs, so it is not surprising that PD-1 and CD28, mainly expressed by T cells, and PD-L1, mainly expressed by macrophages are also more prevalent in cSCCs. Moreover, the authors basically present the same data in different comparisons in Table 2 (and Fig. 3, showing the same thing) vs. Table 4 (and Fig. 5), and the ratios of the different parameters (presented in Table 3 and Fig. 4) also derive from the same data, and it’s not clear what conclusions can be drawn from the latter. It is also not clear why the authors present results only on a subset of the cases involved (and shown in Table 1); for example, results concerning CD28 are presented only for about half of cSCC samples.
- Although (as mentioned above) Table 4 would be expected to show the same results (only in different comparison) as Table 2, the case numbers as well as medians and SDs of the studied parameters in the respective groups are different in many cases (for example, total labeling at the invasive front for PD-1 in BCC has n=69, median 1.12 and SD 2.04 in Table 2 while n=72, median 1.18 and SD 2.09 in Table 4).
- There are also discrepancies in case numbers and/or percentages of T, N and M stage as well as grade between Results text and Table 1. Moreover, in Fig. 6a the correlation coefficient and p values shown in the figure and provided in the figure legend are different.
- It is not described in Materials and Methods whether only immune cells were evaluated for the expression of markers, or tumor cells as well (which is especially important in the case of PD-L1). Probably both, but in this case the title of section 3.2, as well as referring to “immune cell markers” in Discussion is misleading. It is also not described in Methods whether a single core for each location in each sample or multiple cores were included in the TMA.
Minor problems:
- The abbreviation IT (probably meaning immunotherapy) should be defined at first mentioning. “IT significantly depends…” – not immunotherapy itself, but its efficacy depends on the mutational burden.
- Parts of some figure and table legends are shifted and written with larger font letters than the rest of the legends so it is not clear that they belong to the legends.
- In lines 235-236 reference to Figure 3e should be omitted.
- In the legends of Tables 2 and 3 it is written twice that Mann-Whitney U test was used for p value calculation. In Fig. 3 no results of total cell labeling indices are shown so reference to this should be omitted from the figure caption.
- Lines 266-272: “differences towards the numerator” is not clear; it could be simply written that the ratios were higher in BCC or in cSCC.
- The end of the sentence is missing in line 298.
- Figure 5 should be placed before Tables 5 and 6.
- Lines 404-405: increased survival in which diseases? (The reference is about IBD.)
- Since according to the cited results there is minimal difference in the mutational burden between BCC and cSCC, it cannot be stated that BCC has a higher mutational burden (lines 422-423).
-
Author Response
Dear Reviewer,
Please see the attachment.
Sincerely,
Linus Winter

Reviewer 2 Report
Comments and Suggestions for Authors
minor editing is required
Author Response

(The authors gave the same response as above.)

Reviewer 3 Report
Comments and Suggestions for Authors
The study represents a good amount of work to address the research question on comparison of immune checkpoints in BCC and cSCC tumor tissues. The conclusion is supported by the data presented. I have the following suggestions for clarification and /or improvement of the manuscript:
1. Please add description to Figure 1 for the readers. The difference in staining intensity or in the index is not very easily understood from the images.
2. Table 2 and the Figure 3 represent the same data (stroma labeling indices). The box plots perhaps are showing the median and IQR with min and max values. This is appropriate. But in the Table, you are showing SD with median. In general SD is shown along with mean and IQR with the median. Same issue with other Tables 3 and 4.
3. In the result section, it would be better if you first describe the meaning or interpretation of the ratios like PD-1/PD-L1, CD28/CD86 etc (table 3).
4. Please double check the values in Table 4. Should it not exactly match with Table 2?
5. I guess that the Tables 5 & 6 show the same data presented in Figure 6. Figure 6 is sufficient. Even if you prefer to present the data in Table form and not scatter plot, in these two tables, the “*” or “**” are perhaps unnecessary if you present the p-values. I suggest keeping the scatter plots.
6. Previous NGS-based study also indicated that the differential gene expression of PD-L1, LAG3, CTLA4 or HAVCR2 in BCC did not favor potential use of immune checkpoint inhibitor in BCC (PMCID: PMC9688807 ). However the current study utilizes, for the first time, in-depth analysis of immunohistochemistry findings for activating as well as inhibitory factors related to use of immune checkpoint inhibitors in BCC and cSCC. Publication of this manuscript will help the readers.
7. Because of person-to-person variation in expression level, the lack of comparison between tumor and normal tissue is a limitation in this type of tumor BCC to cSCC comparison. From your work, could you suggest any cut-off values for any of the indices that could be used in routine purpose?
8. Please clarify if you meant missing cases by Tx, NX, MX, GX etc in Table 1.
Author Response

(The authors gave the same response as above.)

Round 2
Reviewer 1 Report
Comments and Suggestions for Authors
Most of the errors were corrected, although some remained. I maintain that the new results are incremental (with essentially the same results presented in several tables and figures), therefore the manuscript's merits are not high enough to deserve publication in this journal.
Comments on the Quality of English Language-
Author Response
Dear Reviewer,
Thank you for your efforts to maintain a high standard of publication that we, the authors, share. In response to your feedback, we have moved the former Figure 3 and the former Tables 3 and 4 to the Supplementary Data section. This adjustment ensures that it does not appear that we are presenting the same data in multiple illustrations.
We acknowledge your comments regarding the incremental nature of the results. However, we believe that our study provides a valuable database to address important questions related to immunotherapy in BCC and cSCC.
Regarding the suggestion for minor language editing, we have had the manuscript thoroughly proofread by a native speaker with a doctorate in linguistics. We hope that all language-related concerns have been addressed to your satisfaction.
We hope that these revisions convince you that, as a foundational database for future research, this publication merits inclusion in the journal “cells”.
Sincerely,
Linus Winter